# Cryo-EM structure of the fully-loaded asymmetric anthrax lethal toxin in its heptameric pre-pore state

Claudia Antoni[1,☉], Dennis Quentin[1,☉], Alexander E. Lang[2], Klaus Aktories[2], Christos Gatsogiannis[1], Stefan Raunser[1]*

**1** Department of Structural Biochemistry, Max Planck Institute of Molecular Physiology, Dortmund, Germany, **2** Institute of Experimental and Clinical Pharmacology and Toxicology, Faculty of Medicine, University of Freiburg, Freiburg, Germany

☉ These authors contributed equally to this work.

* stefan.raunser@mpi-dortmund.mpg.de

**Data Availability Statement:** The cryo-EM density maps of the PA7LF3-masked, PA7LF(2+1B), PA7LF(2+1A) and PA7LF(2+1A)', complexes are deposited into the Electron Microscopy Data Bank

## Abstract

Anthrax toxin is the major virulence factor secreted by *Bacillus anthracis*, causing high mortality in humans and other mammals. It consists of a membrane translocase, known as protective antigen (PA), that catalyzes the unfolding of its cytotoxic substrates lethal factor (LF) and edema factor (EF), followed by translocation into the host cell. Substrate recruitment to the heptameric PA pre-pore and subsequent translocation, however, are not well understood. Here, we report three high-resolution cryo-EM structures of the fully-loaded anthrax lethal toxin in its heptameric pre-pore state, which differ in the position and conformation of LFs. The structures reveal that three LFs interact with the heptameric PA and upon binding change their conformation to form a continuous chain of head-to-tail interactions. As a result of the underlying symmetry mismatch, one LF binding site in PA remains unoccupied. Whereas one LF directly interacts with a part of PA called α-clamp, the others do not interact with this region, indicating an intermediate state between toxin assembly and translocation. Interestingly, the interaction of the N-terminal domain with the α-clamp correlates with a higher flexibility in the C-terminal domain of the protein. Based on our data, we propose a model for toxin assembly, in which the relative position of the N-terminal α-helices in the three LFs determines which factor is translocated first.

## Author summary

Anthrax is a life-threatening infectious disease that affects primarily livestock and wild animals, but can also cause high mortality in humans. Due to its suitability as a bioweapon and the search for an antidote, it is important to understand the molecular mechanism of infection of the anthrax pathogen and in particular of the anthrax toxin. Although the process of poisoning by anthrax toxin has been extensively investigated, many details are still missing that are required to understand the mechanism of action in molecular detail. Here, we used single-particle electron cryo microscopy to determine structures of the

with the accession codes EMD-11522, EMD-11523, EMD-11524 and EMD-11525, respectively. Corresponding coordinates for PA7LF3-masked, PA7LF2+1B and PA7LF2+1A have been deposited in the Protein Data Bank under accession number 6ZXJ, 6ZXK and 6ZXL. All relevant data is in the manuscript or has been deposited to the EMDB and PDB and publicly available.

**Funding:** This work was supported by the Max Planck Society (to S.R.) and the European Council under the European Union's Seventh Framework Programme (FP7/ 2007–2013) (grant no. 615984) (to S.R.). D.Q. is a fellow of Fonds der Chemischen Industrie. The funders had no role in study design, data collection and analysis, decision to publish, or preparation of the manuscript.

**Competing interests:** The authors have declared that no competing interests exist.

fully-loaded asymmetric anthrax lethal toxin in its heptameric pre-pore state. The structures reveal that three lethal factors interact with the heptameric protective antigen and upon binding change their conformation to form a continuous chain of head-to-tail interactions. Based on our data, we propose a model for toxin assembly, in which the relative position of the N-terminal region in the three lethal factors determines which factor is translocated first. Our studies provide novel insights into the organization of the anthrax lethal toxin and advance our understanding of toxin assembly and translocation.

## Introduction

Anthrax is a life-threatening infectious disease that affects primarily livestock and wild animals, but can also cause high mortality in humans [1]. During the early and late steps of infection with the Gram-positive bacterium *B. anthracis*, the tripartite anthrax toxin is secreted as major virulence factor in order to kill host immune cells such as macrophages or neutrophils [2,3]. Like other AB-type toxins, it is composed of a surface binding/translocation moiety, the protective antigen (PA, 83 kDa), and two cytotoxic subunits, lethal factor (LF, 90 kDa) and edema factor (EF, 93 kDa) [4,5].

To execute their toxicity, both the zinc-dependent metalloproteinase LF and/or the adenylate cyclase EF need to enter the host cytoplasm [6,7]. For that purpose, PA monomers first attach to the cell surface through binding to one of the two known membrane receptors, capillary morphogenesis gene 2 (CMG-2) and tumor endothelial marker 8 (TEM8) [8,9]. After cleavage by furin-like proteases, the truncated 63 kDa-sized PA monomer oligomerizes either into homo-heptamers ($PA_7$) or homo-octamers ($PA_8$) [10–12]. These ring-shaped oligomers, enriched in lipid raft regions, are in a pre-pore conformation as they do not penetrate the host membrane [13]. Due to the enhanced stability of $PA_8$ under diverse physiological conditions, it is proposed that the octameric form could circulate in the blood to reach and exert toxicity even in distant tissues [14]. This suggests that both oligomeric forms play an important role in intoxication, endowing *B. anthracis* with greater versatility against its host.

In the next step, the holotoxin is assembled by recruiting LFs/EFs. While $PA_8$ can bind up to four factors, only three of them can simultaneously bind to $PA_7$. Both enzymatic substrates bind to the upper rim of the PA oligomer via their N-terminal domains in a competitive manner [15]. Loaded complexes are then endocytosed [16,17], followed by a conformational change from the pre-pore to pore state which is triggered by the low pH in the endosome [18]. The central feature of the pore state is an 18 nm long 14-stranded β-barrel that spans the endosomal membrane with its narrowest point in the channel lumen being ~6 Å in width [19]. To pass through this hydrophobic restriction, called Φ-clamp, the substrate needs to be unfolded prior to translocation [20].

Structural and functional studies on the pre-pore PA octamer bound to four LFs revealed that an amphipathic cleft between two adjacent PA protomers, termed α-clamp by Krantz and coworkers, assists in the unfolding process [21]. More specifically, the first α-helix and β-strand (α1-β1) of LF almost completely unfold and change their position respective to the rest of the protein when interacting with the α-clamp of the PA oligomer [21]. After transition into the pore state, the unidirectional translocation of LF is driven by a proton-motive force, comprising the proton gradient between the two compartments and the membrane potential. It is thought that the acidic pH present in the endosome destabilizes the LF and thus promotes unfolding of its N-terminus [22]. Ultimately, it is believed that the translocation follows a 'charge-dependent Brownian ratchet' mechanism [23]. The required unfolding and refolding

of translocated enzymes is facilitated *in vivo* by chaperones, but can occur *in vitro* without the need of accessory proteins [24,25].

Crystallographic studies provided us with structural insights pertinent to the molecular action of the anthrax toxin. This includes structures of the individual complex subunits such as LF, EF and the PA pre-pore in both, its heptameric and octameric form [12,26–29]. The PA monomer was also co-crystallized with its receptor CMG-2, delineating the surface attachment to the host cell in molecular detail. More recently, the elusive pore state of $PA_7$ was determined by electron cryo-microscopy (cryo-EM) in which Jiang *et al.* made use of an elegant on-grid pore induction approach [30].

In contrast, high resolution information on holotoxin complexes is rather scarce. The only obtained crystallographic structure is the aforementioned $PA_8$ pre-pore in complex with four LFs [21]. In this structure, however, the C-terminal domain of LF is absent. Unlike $PA_8$, loaded $PA_7$ was mainly studied by cryo-EM [31–35], presumably because its asymmetry impeded crystallization efforts. Earlier this year, the $PA_7$ pore state decorated with a single LF molecule and with up to two EF molecules was determined, in which it was shown that EF undergoes a large conformational rearrangement as opposed to LF [36]. However, cryo-EM studies of the loaded heptameric pre-pore were so far limited to a resolution of ~16 Å [31,32,34]. In addition, the number of LFs bound to $PA_7$ varied between one and three in these structures.

Here, we present three cryo-EM structures of the fully loaded anthrax lethal toxin in the heptameric pre-pore state ($PA_7LF_3$), in which three LF molecules are bound to the rim of the $PA_7$ ring, forming a continuous chain of head-to-tail interactions. The position and conformation of the LFs, however, varies between the structures. Unexpectedly, only one of three LFs interacts with the α-clamp of PA, adopting the "open" conformation as reported in the $PA_8LF_4$ structure [21]. Since we could neither observe a similar interaction for the other two LFs, nor them being in the "closed" conformation, we propose that they adopt an "intermediate" state between holotoxin assembly and translocation. Our findings allow us to propose a model for anthrax lethal toxin assembly, in which the LF translocation sequence is dictated by the position of the N-terminal α-helix of the LFs.

## Results

### Structure of the fully-loaded anthrax lethal toxin in the heptameric pre-pore state

To ensure that our purified and reconstituted $PA_7$ complexes (Materials and Methods) are indeed intact, we tested their membrane insertion capacity by reconstituting them in liposomes or nanodiscs (S1 Fig). We then evaluated different molar ratios of $LF:PA_7$ and only obtained fully-loaded anthrax lethal toxin ($PA_7LF_3$) when using a 10:1 molar ratio as judged by size exclusion chromatography (S2A and S2B Fig).

We then determined the structure of the $PA_7LF_3$ pre-pore complex by single particle cryo-EM to an average resolution of 3.5 Å. However, in this initial 3D reconstruction the densities corresponding to LF represented a mixture of assemblies and were partly unassignable (S3 Fig). This can be either due to the symmetry mismatch that emerges when three lethal factors bind simultaneously to $PA_7$ or to possible different conformations of the individual LFs bound to $PA_7$. To address these points, we established an image processing workflow that includes sequential 3-D classifications and rotation of classes (S3 Fig). This resulted in three reconstructions with resolutions of 3.8 Å, 4.2 Å and 4.3 Å that differed in the position of the third LF bound to $PA_7$ (1A and Fig 1B, S2, S3 and S4 Figs) and the conformation of LF (Fig 1C, S2, S3 and S4 Figs). The densities corresponding to the lethal factors in the 4.3 Å structure were not resolved well enough to allow the fitting of an atomic model (S2F, S2I and S4B Figs).

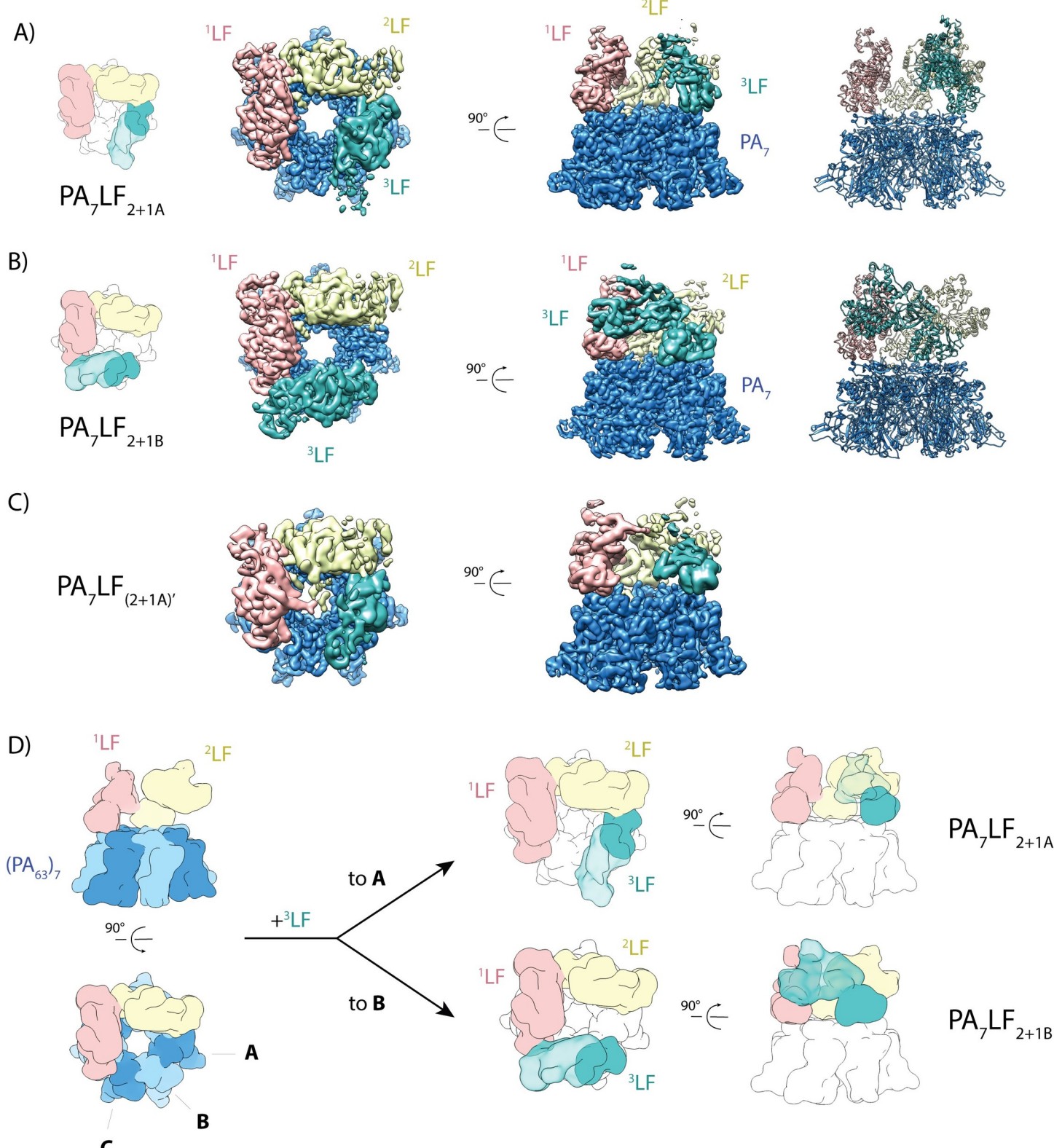

**Fig 1. Cryo-EM structures of the PA$_7$LF$_3$ complexes.** (A) Top view and side view of the color-coded segmented cryo-EM density map of PA$_7$LF$_{2+1A}$, with PA$_7$ in blue, $^1$LF in pink, $^2$LF in gold and $^3$LF in cyan. The naming of the three LFs is based on their respective local resolution with $^1$LF being best-resolved, while $^3$LF showed the weakest density in all reconstructions. Three lethal factors bind to the PA$_7$ ring and form a continuous chain of head-to-tail interactions. Schematic representation is

shown on the left, corresponding atomic model on the right. (B) Same as in (A) for the $PA_7LF_{2+1B}$ complex. (C) Same as in (A) for the $PA_7LF_{(2+1A)'}$ complex. Notably, two LFs interact in their peripheral region (C-terminal domain) with each other close to the central axis. Segmented maps are shown at different thresholds for visualization. (D) Schematic representation of the last step in $PA_7LF_3$ toxin assembly, in which the third lethal factor can bind to one of two empty PA sites, resulting in two different complexes, $PA_7LF_{2+1A}$ and $PA_7LF_{2+1B}$. Top and side views are shown, with the same color code as in (A), except that PA protomers alternate in light and dark blue.

Therefore, we proceeded with the remaining two structures, combined the two particle stacks and masked out the density of the third LF to improve the resolution of the rest of the complex to 3.5 Å (S2H, S3 and S4D Figs). Using a combination of the maps, we then build atomic models for the 3.8 Å and 4.2 Å reconstructions (Fig 1A and 1B, S1 Table).

The structures reveal that $PA_7$ forms a seven-fold symmetric ring structure with a ~25 Å wide central opening. With the exception of a few poorly resolved loop regions in the periphery of $PA_7$, our structures almost perfectly superimpose with the crystal structure of the $PA_7$ pre-pore (PDB:1TZO; RMSD of 0.92 Å) [26] (S5A Fig), indicating that the binding of LF does not induce conformational changes in $PA_7$. This is in contrast to Ren et al. who suggested that LF binding results in a distortion of the symmetric $PA_7$ ring, thereby facilitating the passage of cargo through the enlarged lumen [31,37]. Noteworthy, the 2β2-2β3 loop region (residues 300–323) which is implicated in pore formation was not resolved in our map. This indicates a high flexibility of this loop, which is in line with previous MD simulations [38].

In all $PA_7LF_3$ structures, the three LFs sit on top of $PA_7$. The densities corresponding to the LFs show a resolution gradient from the central N-terminal domain which is resolved best to the peripheral C-terminal domain (S4A–S4D Fig). This indicates that this region is quite flexible compared to the rest of the toxin complex. We named the three LFs based on their respective resolution with [1]LF being best resolved and [3]LF having the weakest density.

The LFs do not only interact with $PA_7$ but also form a continuous chain of head-to-tail interactions with each other. Binding of LF to a single PA protomer is mediated via the N-terminal domain of LF, orienting its bulky C-terminal domain such that the adjacent PA protomer is not accessible for binding. In this way a single lethal factor *de facto* occupies two of the seven binding sites of $PA_7$. In the chain of LFs, the C-terminal domain of the anterior LF binds to the N-terminal domain of the following one, creating a directionality in the complex (Fig 1D). Consequently, if two LFs are bound, three free PA binding sites are available, of which only two can potentially be occupied due to steric clashes (Fig 1D). This results in the two complexes $PA_7LF_{2+1A}$ and $PA_7LF_{2+1B}$, that differ in the binding position of the third LF (Fig 1). Since each LF occupies two potential binding sites in these structures, this leads to a symmetry mismatch and leaves one PA unoccupied.

## Crucial interactions in the $PA_7LF_3$ complex

LF and PA interact mainly via a large planar interface at which domain I of LF interacts with the LF/EF binding sites of two adjacent PAs (Fig 2, S5B Fig). The interaction with LF is mediated primarily by one PA, whereas the contribution of the second PA is only minor (Fig 2A). The LF-PA interface is well resolved for all LFs and almost identical in the different structures (Fig 2A, S5A and S5C Fig). The interaction is primarily mediated by an extensive hydrophobic core that is further surrounded by electrostatic interactions (Fig 2A). The interface in our structure is very similar to the one previously described for $PA_8LF_4$ [21]. There, the second LF-PA interface is formed by the N-terminal α-helix of LF that interacts with the α-clamp located at the interface of two PAs. This "open" conformation differs from the "closed" conformation of this region as observed in the structure of the unbound LF [27]. When comparing the LFs in our structure with that of the unbound LF [27], we observed that the C-terminal domain of the LFs in $PA_7LF_3$ is rotated in relation to the N-terminal domain, bringing them

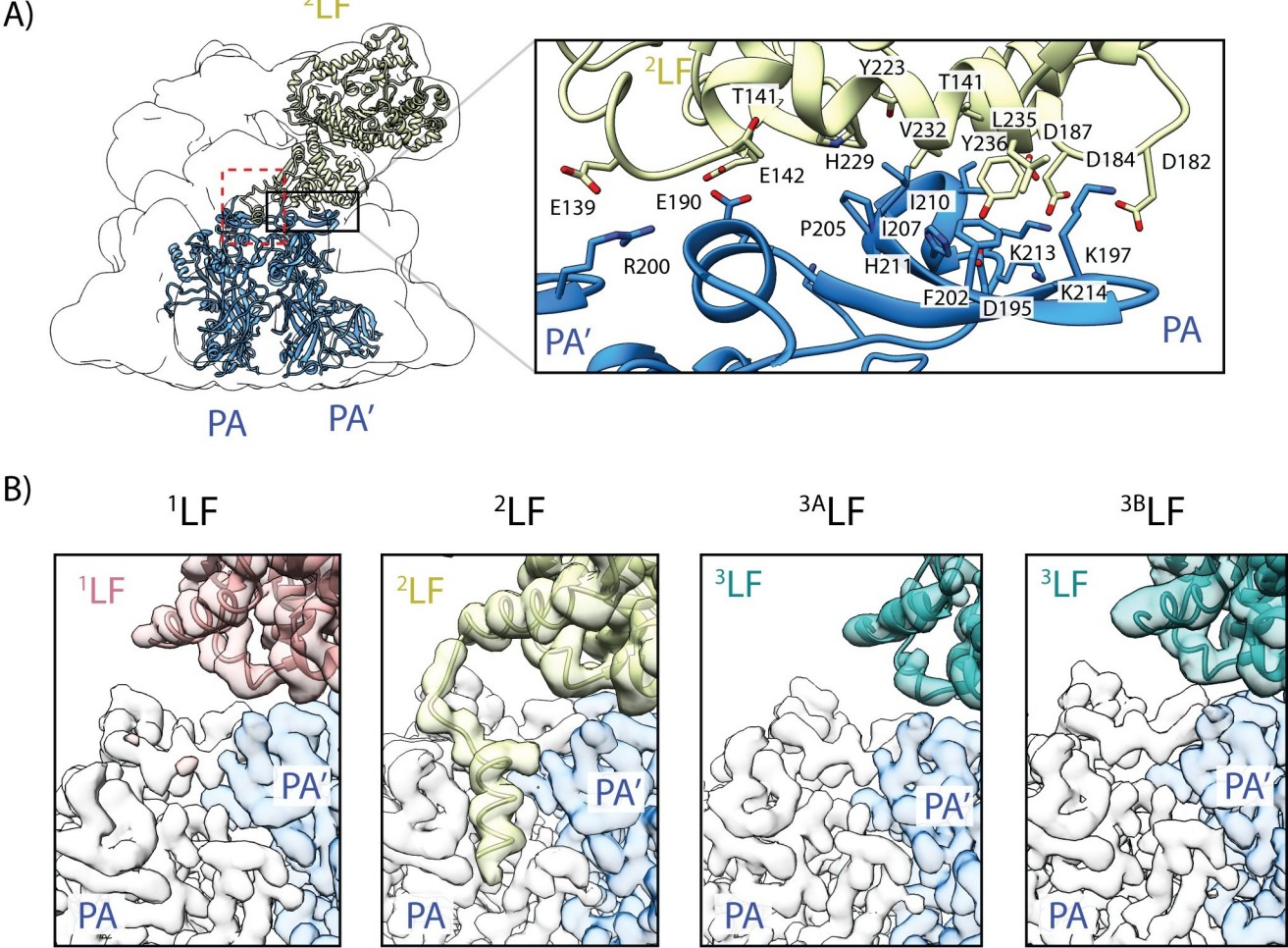

**Fig 2. Interfaces between lethal factor and protective antigen.** (A) The N-terminal domain of LF mediates binding to two adjacent PA molecules, forming a large planar interface. The positions of [2]LF (gold), PA and PA'(blue) are shown relative to the overall shape of the complex that is represented as transparent, low-pass filtered volume. A black square indicates the interaction interface between all three molecules. The inset shows a close-up of the interaction regions, with contributing residues labeled. They form a central hydrophobic core, that is surrounded by electrostatic interactions. (B) The second LF-PA interface, indicated by a dashed square in (A), is formed by the N-terminal α-helix of LF, interacting with the α-clamp region, located between two adjacent PA molecules. The four panels depict a close-up of this region for the three different LFs ([3]LF can adopt two different positions, i.e. the PA7LF2+1A or PA7LF2+1B complex) with half-transparent densities shown for PA (white), PA' (light blue) and the LF ([1]LF—pink; [2]LF—gold; [3]LF–cyan). Notably, only [2]LF interacts with the α-clamp.

closer together (Fig 3, S1 Movie). However, we only found that the N-terminal region of one LF ([2]LF) resides in the α-clamp, adopting the "open" conformation as described for PA8LF4 [21]. In the other LFs ([1]LF, [3]LF), this region is flexible and not interacting with the α-clamp (Fig 2B). A steric clash between the loop region (residues 576–579) of [1]LF and α1-β1 of [2]LF (Fig 4A) prevents the N-terminal α-helix of [2]LF from remaining in the "closed" conformation. Since [1]LF and [3]LF neither take the "open", nor the "closed" conformation, we propose that they reside in an "intermediate" conformational state between toxin assembly and translocation. We further hypothesize that [2]LF is the first of the three lethal factors that is unfolded by PA7 and is also the first one to be translocated.

As described above, the LFs interact via their N- and C-terminal regions. In two of our structures, PA7LF2+1A and PA7LF2+1B, two LFs only interact at one position which is located next to the major LF-PA interface. In the third structure, which we designate as PA7LF(2+1A)',

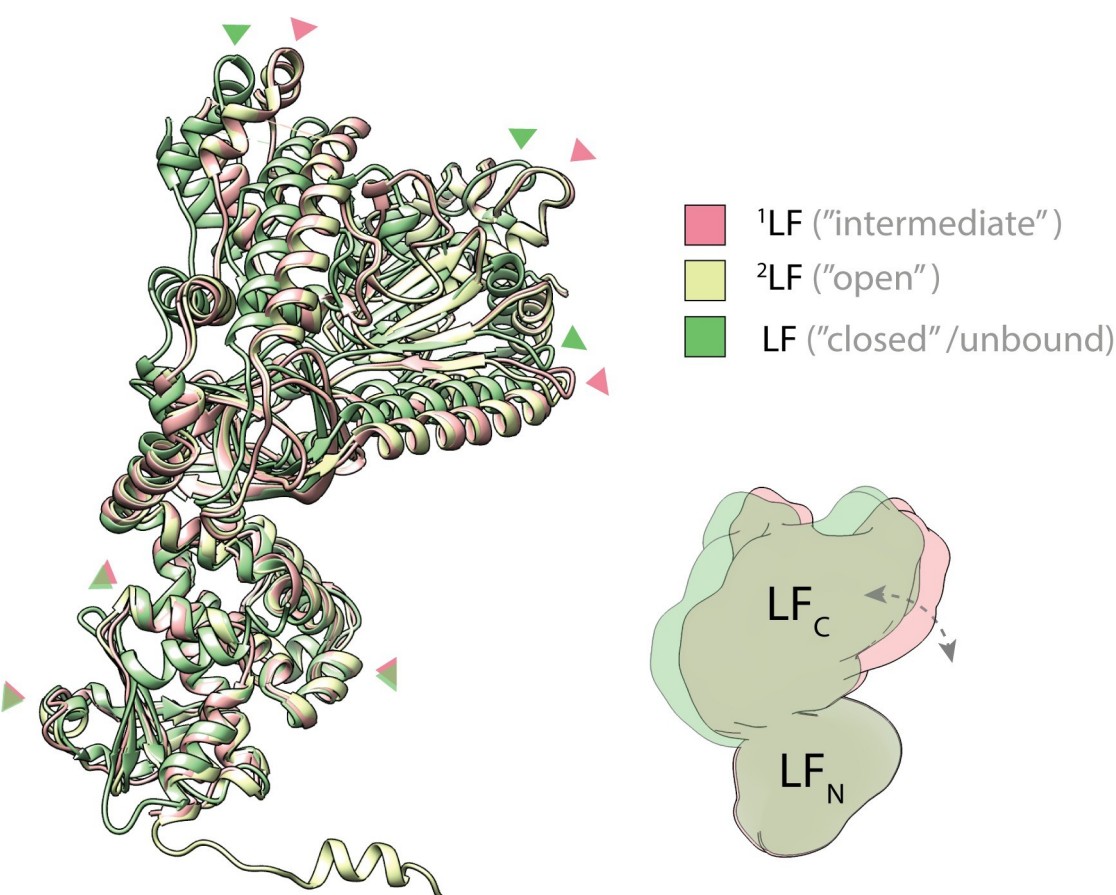

**Fig 3. Conformational change of LF upon PA binding.** Superposition of [1]LF (green, "intermediate" conformation), [2]LF (yellow, "open" conformation) and unbound LF (green, "closed" conformation, PDB: 1J7N), aligned via their N-terminal domains. Red and green arrows indicate similar positions in [1]LF and unbound LF, respectively. When compared with the crystal structure of the unbound lethal factor (PDB: 1J7N), the three LFs undergo a conformational change upon interaction with PA$_7$. The C-terminal domain rotates with respect to the N-terminal domain such that the LFs come closer to form a continuous chain of head-to-tail interactions. A schematic representation illustrates the rotation of the C-terminal domain that occurs between unbound (green) and bound (red) LF conformation. See also S1 Movie.

two LFs likely interact with each other also via their C-terminal region close to the central axis of the complex (Fig 5, S2 Movie). However, the position of this additional interface that we observe in our PA$_7$LF$_{(2+1A)}$, structure differs from the putative interface, that has been proposed by Fabre et al, located further away from the central axis [34]. The main [2]LF-[1]LF interface, that is present in all three structures, consists of the helix-loop region (residues 572–579) of the first lethal factor ([1]LF) which forms a relatively small interaction surface with the helix-helix-β-sheet motif (residues 52–84) of the adjacent lethal factor ([2]LF) (Fig 4B). Residues L63, L71 and I81 of [2]LF form a central hydrophobic cavity that interacts with Y579 of [1]LF. In the β-sheet region of [2]LF, we identified a potential backbone-backbone hydrogen bond between K578 and I81 of [1]LF. In addition, P577 forms a hydrophobic interaction with Y82, which is further stabilized by H91. K572, being located on the α-helix next to the loop region in [1]LF, could potentially form a salt bridge interaction with E52 or D85 of [2]LF. Together these interactions mediate the binding between two LFs. Although the local resolution at the [2]LF-[3A]LF and [3B]LF-[1]LF interfaces does not allow the fitting of side chains (S4A–S4D Fig), we could flexibly fit in the structures of [1]LF and [2]LF at this position. Since all structures are almost identical at

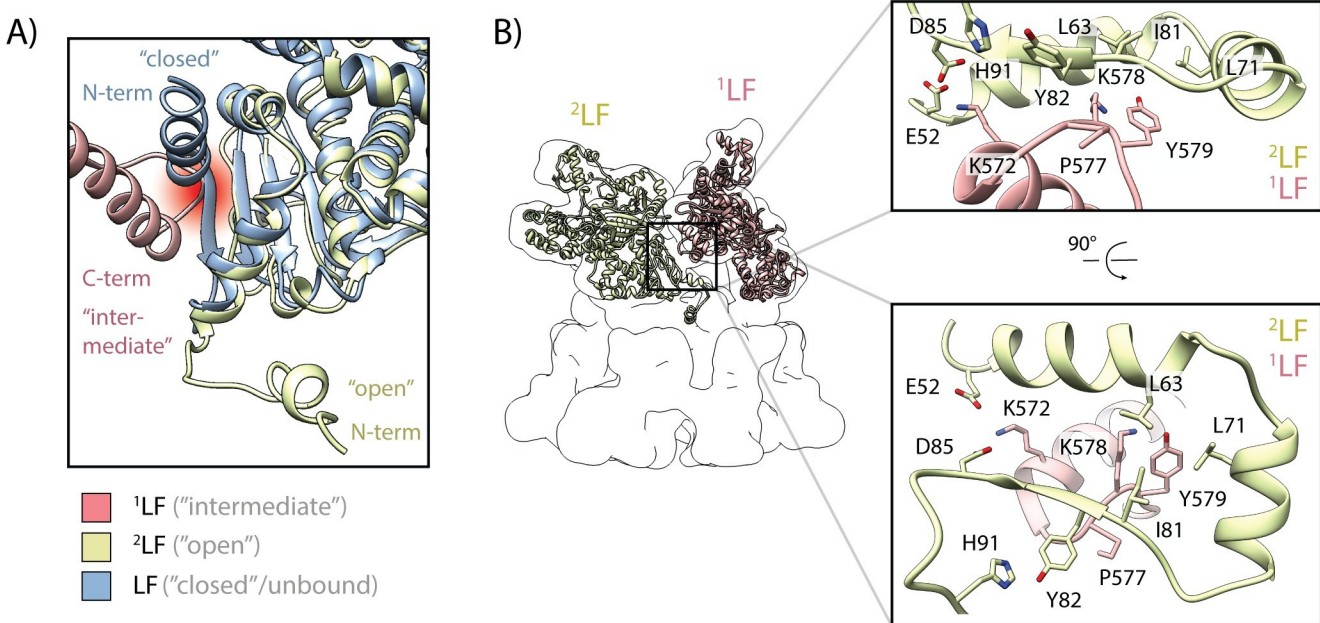

**Fig 4. Molecular interface between two lethal factors.** (**A**) Potential steric clash between the C-terminal domain of LF (red) and the N-terminal domain of an adjacent LF when it adopts the "closed" conformation (blue). The clash is highlighted as fading red spot in the background. In contrast, the "open" conformation (gold), i.e. the N-terminal α-helix interacts with the α-clamp region of PA, does not result in a steric clash. The N-terminal region of the red LF, which adopts an "intermediate" conformation, is not shown in this view. (**B**) A relatively small interaction interface mediates binding of the C-terminal domain of LF to the N-terminal domain of an adjacent LF. The positions of [1]LF (pink) and [2]LF (gold) are shown relative to the overall shape of the complex that is represented as transparent, low-pass filtered volume. A black square indicates the interaction interface between the two LFs. Insets show close-ups of the interacting regions in different orientations, with contributing residues labeled.

backbone level (RMSD of 0.84 Å and 0.96 Å) (S5F Fig), we expect them to exhibit a similar network of interactions. Both interfaces, LF-PA and LF-LF that we describe here limit the freedom of movement mainly in the N-terminal region of LF, but still allow a certain level of flexibility in the rest of the protein.

In all structures, the LFs show a gradient in flexibility (Fig 1A–1C, S4A–S4D Fig). This was previously not observed at lower resolution [34]. [1]LF is resolved best in all structures, followed by [2]LF and [3]LF has the weakest density in all reconstructions. Since the N-terminal domain is well resolved in all LFs, this cannot be due to a varying occupancy of the binding sites, but must stem from a flexibility of the C-terminal domain. As expected, all free C-terminal domains, i.e. those that are not stabilized by an adjacent LF are more flexible than those with a binding partner. However, there is one exception, namely [2]LF. In this case, the C-terminal domain is always flexible, independent of a stabilizing binding partner. Interestingly, [2]LF is also the only lethal factor where the N-terminal α-helix of LF is ordered and resides in the α-clamp, suggesting that this interaction results in a destabilization of the C-terminal domain of the molecule. This is in line with a recently reported structure of the $PA_7LF_1$ pore state where the C-terminal domain of the single LF bound was not resolved while the N-terminal α-helix is also bound to the α-clamp [36].

## Discussion

We determined three structures of the fully-loaded heptameric anthrax lethal toxin complex, which differ in the position and conformations of the bound LFs. Due to a symmetry mismatch, three LFs occupy six binding sites of the heptameric $PA_7$ complex, leaving one PA site

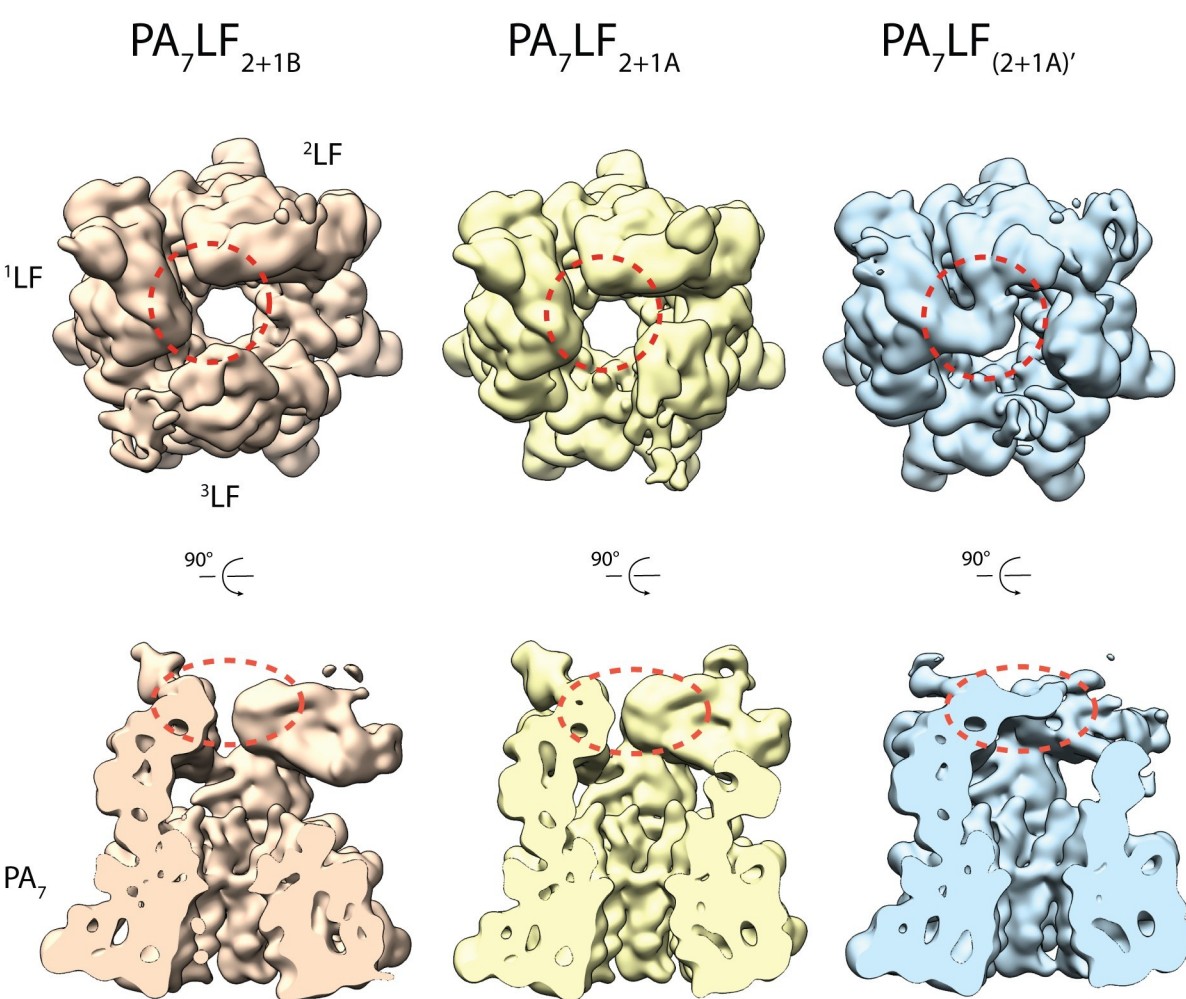

**Fig 5. LFs can interact via their C-terminal domain.** Top and side views of the low-passed filtered maps of the three PA$_7$LF$_3$ complexes, with PA$_7$LF$_{2+1B}$ in orange, PA$_7$LF$_{2+1A}$ in yellow and PA$_7$LF$_{(2+1A)'}$ in light blue. Volumes are shown at the same threshold. While the three LFs interact in all structures via their N- and C-terminal domains in a head-to-tail manner, an additional interface was identified in the PA$_7$LF$_{(2+1A)'}$ reconstruction. Here, the C-terminal domains of $^1$LF and $^2$LF, interact with each other close to the central axis of the PA$_7$LF$_3$ complex. This region is highlighted by dashed red circles. See also S2 Movie.

empty. Compared to the "closed" state as observed in the crystal structure of LF [27], the C-terminal domain of the LFs in PA$_7$LF$_3$ is rotated respective to the N-terminal domain. However, only $^2$LF adopts the "open" conformation which was reported for the structure of PA$_8$LF$_4$ [21], i.e. the N-terminal α-helix interacts with the α-clamp of PA. $^1$LF and $^3$LF do not show this interaction, but can also not be in the "closed" conformation because of a steric clash with an adjacent LF. We therefore propose that they are in an "intermediate" state between toxin assembly and translocation.

Loading of PA$_7$ with one or more substrates is crucial for the anthrax toxin to mediate toxicity during infection. The high binding affinity of LF and EF to PA$_7$ enables the formation of a stable toxin complex, which is required to withstand the rather hostile environment present within the host during the various steps of the infection process. A dissociation constant (K$_D$) in the range of 1–2 nM has been determined by both, surface plasmon resonance (SPR) and binding studies using radiolabeled substrate with receptor-bound PA on the surface of L6 cells [39]. Head-to-tail binding of LFs as observed in our fully loaded PA$_7$LF$_3$ structures further

contributes to the overall stability of the complex. This increased stability might be important during the internalization step and subsequent endosomal trafficking, before finally reaching its cytosolic target. It might reduce the susceptibility towards proteases and the probability to prematurely transition into the pore-state.

Studies addressing the saturation state of $PA_7$ (and $PA_8$) with LF/EF ligands in vivo are rather limited. While one study of cutaneous infection in immune-competent mice has focused on the level of both substrates and their respective ratio over the course of infection, the number of bound substrates to $PA_7$ or $PA_8$ has not been determined [40]. As has been suggested before, it is likely that this number can vary depending on the stage of the infection process, as well as the location within the host. Interestingly, mass spectrometry data of toxin assembly in vitro did not identify $PA_7$ or $PA_8$ ring intermediates decorated with less than three or four LFs, respectively [12]. This observation, however, could be a consequence of the two different assembly routes that will be discussed later. Furthermore, it has been shown by Zhang et al. that the number of ligands bound to $PA_7$ did not affect the translocation efficiency [41]. By varying either the ligand concentration or the available ligand binding sites in $PA_7$, they excluded both positive and negative cooperativity in the translocation process of radiolabeled LF. These findings suggest that LF and EF ligands are translocated independently of the number of bound ligands.

In our structures, we observed that two LFs adopt a so called "intermediate" state, in which their N-terminal α-helix neither interacts with the α-clamp of PA, nor adopts the "closed" conformation [27]. Why has this state not been observed in the crystal structure of $PA_8LF_4$? It could have been missed due to averaging of the asymmetric unit of the $PA_8LF_4$ crystals, which is composed of two PAs and one LF. Another possibility is that compared to $PA_7$, the $PA_8$ pre-pore provides more space for the N-termini of the LFs to arrange in the "open" conformation in comparison to the $PA_7$ pre-pore. However, if all LFs were indeed in a "ready-to-be-translocated position" which LF would then be translocated first through the narrow PA pore that only allows the passage of a single unfolded LF at a time? The process could in principle be stochastic, but our $PA_7LF_3$ structures offer an alternative explanation.

Already based on the low-resolution structure of the $PA_7LF_3$ pre-pore [34], it has been suggested that the order of translocation is non-stochastic and that the first LF, whose N-terminal domain is not interacting with an adjacent LF, is translocated first. After the translocation of this factor, the second LF would be released from the inhibitory bond of the first LF and then be translocated and so on [34]. However, our data indicate that such an order in which the translocation of one LF is dependent on the translocation of an adjacent factor is rather unlikely.

Although we can as well only speculate about the exact order of translocation, based on our cryo-EM structures, two alternative scenarios are conceivable: In the first one, the factor in the "open" state, [2]LF, is translocated first, followed by [1]LF or [3]LF which are in the "intermediate" state (Fig 6). The second possibility would be that [1]LF and [3]LF are translocated before [2]LF (Fig 6). Due to the different arrangements in the complexes, both alternatives exclude a dependency on a neighboring factor for LF translocation.

While we cannot exclude the second scenario, we think that the first one is more likely. Being in the "open" conformation, the N-terminal α-helix of [2]LF interacts with the α-clamp of PA. The α-clamp is known to unfold polypeptides in a sequence-independent manner. The current theory is that this crucial region formed by two adjacent PA molecules first stabilizes unfolding intermediates, and introduces mechanical strain before the unfolded structure is fed further down the central pore [21]. In this way the α-clamp would facilitate the rapid unfolding of the entire [2]LF molecule upon transition into the pore state. We therefore believe that [2]LF is translocated before [1]LF and [3]LF. The higher flexibility in the C-terminal domain of [2]LF in the

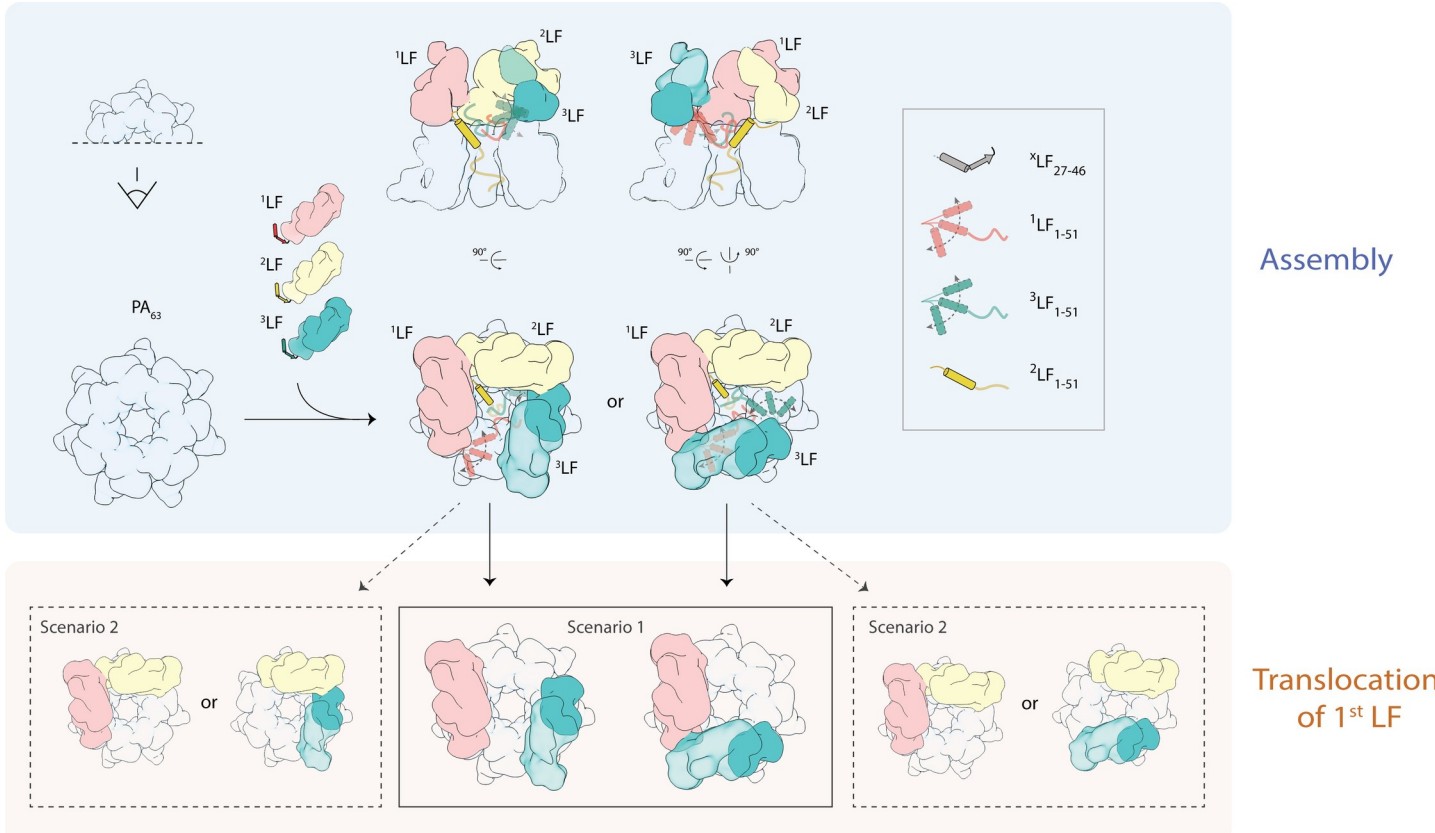

**Fig 6. Model for PA$_7$LF$_3$ assembly and translocation.** After anthrax-receptor-mediated PA$_7$ formation on the surface of the host cell, three LFs bind to PA$_7$ to assemble into the fully loaded anthrax lethal toxin. As a consequence of the symmetry mismatch between PA$_7$ and three LFs, the third LF can attach to two different PA sites, resulting in two distinct PA$_7$LF$_3$ complexes. In both arrangements, the three LFs form a continuous chain of head-to-tail interactions. One LF, namely $^2$LF, undergoes a conformational change from the "closed" state as observed in the unbound LF structure to an "open" state, which is characterized by an interaction of its N-terminal α-helix with the α-clamp region of PA. The other two LFs, $^1$LF and $^3$LF, adopt an "intermediate" state, i.e. being neither in an "open" nor "closed" conformation. Two different scenarios for the translocation of the 1$^{st}$ LF after transitioning into the pore state are conceivable: In scenario 1, $^2$LF, being in the "open" state is translocated first, whereas either $^1$LF or $^3$LF are translocated first in scenario 2.

presence of potentially stabilizing neighboring LFs suggests that the interaction of the N-terminal domain with the α-clamp results in a destabilization of the molecule. This in turn lowers the energy barrier for the unfolding of the entire LF molecule and further supports the assumption that $^2$LF is translocated first. Once $^2$LF is translocated, either $^1$LF or $^3$LF can follow. As these two LFs both adopt an "intermediate" conformation in our structures, we cannot predict which LF is translocated next.

Currently, two possible routes for toxin assembly have become established within the anthrax field: (I) Anthrax-receptor (ATR)-mediated assembly on the cell surface and (II) ATR-independent assembly in solution, i.e. the bloodstream. It is important to note that both routes are not mutually exclusive. In route I, ATR first recruits full-length PA to form PA$_7$ and PA$_8$, where PA is cleaved into its mature form, followed by LF/EF substrate recruitment. In vitro mass spectrometry data indicated that the PA oligomerization process occurs via even-numbered PA$_2$ and PA$_4$ intermediates, being consistent with a proposed dimerization of ATR [42]. In route II, binding of LF/EF to two PA molecules facilitates the assembly of both, the heptameric and octameric PA ring. Similar to route I, assembly route II is populated with dimeric and tetrameric PA intermediates, bound to one and two LFs, respectively [12]. As we

have pre-assembled the $PA_7$ ring before LF was added, our assembly approach resembles rather route I. We believe that $^2$LF is not only the first LF being translocated, but might also be the first one that binds to $PA_7$ during toxin assembly (Fig 6). Upon binding to PA, $^2$LF undergoes a conformational change from the "closed" to the "open" state. The two subsequent LFs adopt an "intermediate" state instead of transitioning into the "open" conformation. Since the third LF can attach to two different PA sites, this results in two different complexes (Fig 6). In this way, the assembled toxin has three LFs bound to $PA_7$ with two in an "intermediate" and one in the "open" conformation (Fig 6).

In summary, our high-resolution cryo-EM structures provide us with novel insights into the organization of the fully-loaded heptameric anthrax lethal toxin and thus advance our understanding of toxin assembly and translocation.

## Materials and methods

### Protein expression and purification

Protective antigen (PA) from *Bacillus anthracis* was cloned into a pET19b vector (Novagen), resulting in a N-terminal $His_{10}$-tag fusion construct. *E. coli* BL21(DE3) were transformed with the pET19b::$His_{10}$-PA plasmid and expression was induced immediately after transformation by the addition of 75 μM IPTG. Following incubation at 28˚C for 24 h in LB medium, cells were pelleted, resuspended in lysis-buffer (20 mM Tris-HCl pH 8.5, 300 mM NaCl, 500 μM EDTA, 5 μg/ml DNAse, 1 mg/ml Lysozym plus Protease inhibitor cOmplete tablets from Sigma Aldrich) and lysed by sonication. Soluble proteins were separated from cell fragments by ultracentrifugation (15,000 rpm, 45 min, 4˚ C) and loaded onto Ni-IDA beads (Cube Biotech). After several washing steps, the protein was eluted with elution buffer (500 mM imidazole, 20 mM Tris-HCl pH 8.5, 500 mM NaCl, 1 mM EDTA). Protein-containing fractions were pooled and dialyzed against buffer containing 50 mM Tris-HCl pH 8.5, 150 mM NaCl, 1 mM EDTA. Subsequently, the sample was further purified using anion-exchange Mono Q (GE Healthcare) with a no-salt buffer (20 mM Tris-HCl pH 8.5) and high-salt buffer (20 mM Tris-HCl pH 8.5, 1M NaCl), applying a gradient from 0 to 40%. Next, oligomerization of PA was induced by addition of trypsin (1 μg enzyme for each mg of PA), followed by incubation on ice for 30 min. Upon addition of double molar excess of trypsin inhibitor (Sigma Aldrich), $PA_7$ was further purified by size exclusion chromatography using a Superdex 200 column (GE Healthcare). Lyophilized LF (List Biological Lab. Inc., Lot#1692A1B) were resuspended in water according to the manufacturer's manual and mixed with $PA_7$ in a molar ration of 10:1. Ultimately, loaded complexes were further purified in a final size exclusion chromatography step (20 mM Tris-HCl pH 8.5, 150 mM NaCl) using a Superdex 200i column (GE Healthcare), before being used in down-stream applications.

### Reconstitution of $PA_7$ in lipid-mimetic systems

For nanodisc insertion, Ni-NTA column material was first washed with $ddH_2O$ and subsequently equilibrated with buffer A (50 mM NaCl, 20 mM Tris-HCl–pH 8.5, 0.05% Octyl β-D-glucopyranoside (w/v)). In the next step, 500 μL of 0.2 μM $PA_7$ in the pre-pore state was added and incubated for 25 min at room temperature. An additional washing step with buffer A was performed to remove unbound $PA_7$ pre-pore, followed by a 5 min incubation step with 1 M urea at 37˚C and another wash with buffer A. MSP1D1:POPC:sodium cholate ratio and preparation was done according to Akkaladevi et al [33]. After dialysis (MWCO of 12-14k) for 24 to 72 h against buffer B (50 mM NaCl, 20 mM Tris-HCl pH 7.5), excess of nanodiscs was collected from five washing steps with 500 μL of buffer B. To elute $PA_7$ pores inserted into nanodiscs, column material was incubated for 10 min on ice in buffer C (500 mM NaCl, 50 mM

Tris-HCl pH 7.5, 50 mM imidazole). The eluted sample was concentrated and subsequently used for negative staining EM.

For the preparation of pre-formed liposomes, POPC was initially solubilized in 5% Octyl β-D-glucopyranoside. Solubilized lipids were dialyzed (MWCO: 12–14 k) for 8–12 h at 4°C against buffer A and subsequently $PA_7$ pre-pores were added to the lipids in a 1: 10 molar ratio. Following 24–72 h dialysis (MWCO:12–14 k) against buffer D (50 mM NaCl 50 mM NaOAc, pH 5.0), samples were used for negative staining EM.

## Negative-stain electron microscopy

Complex purity and integrity were assessed by negative stain electron microscopy prior to cryo-EM grid preparation and image acquisition. For negative stain, 4 μl of purified $PA_7LF_3$ complex at a concentration of ~0.04 mg/ml was applied onto a freshly glow discharged carbon-coated copper grid (Agar Scientific; G400C) and incubated for 45 s. Subsequently, excess liquid was blotted away with Whatman no. 4 filter papers. The sample was stained with 0.8% (w/v) uranyl acetate (Sigma Aldrich). Micrographs were recorded manually using a JEOL JEM-1400 TEM, operated at an acceleration voltage of 120 kV, equipped with a 4,000 × 4,000 CMOS detector F416 (TVIPS) and a pixel size of 1.84 Å/px.

## Sample vitrification

For Cryo-EM sample preparation, 4 μl of purified $PA_7LF_3$ at a concentration of ~0.06 mg/ml was applied onto freshly glow discharged grids (Quantifoil R 1.2/1.3 holey carbon with a 2 nm additional carbon support) and incubated for 45 s. Subsequently, grids were blotted automatically (3 s blotting time) and plunged into liquid ethane using a CryoPlunge3 (Gatan) at a humidity of ~ 95%. Grid quality was screened before data collection using a JEOL JEM-1400 TEM electron microscope (same settings as for negative-stain electron microscopy) or with an Arctica microscope (FEI), operated at 200 kV. Grids were kept in liquid nitrogen for long-term storage.

## Cryo-EM data acquisition

Cryo-EM data sets of $PA_7LF_3$ were collected on a Titan Krios transmission electron microscope (FEI) equipped with a high-brightened field-emission gun (XFEG), operated at an acceleration voltage of 300 kV. Micrographs were recorded on a K2 direct electron detector (Gatan) at 130,000 x magnification in counting mode, corresponding to a pixel size of 1.07 Å. 40 frames taken at intervals of 375 ms (1.86 e⁻/Å²) were collected during each exposure, resulting in a total exposure time of 15 s and total electron dose of 74.4 e⁻/Å². Using the automated data collection software EPU (FEI), a total of 5238 micrographs with a defocus range between -1.2 and -2.6 μm was automatically collected.

## Image processing and 3-D reconstruction

Micrographs of the dataset were inspected visually and ones with extensive ice contamination or high drift were discarded. Next, frames were aligned and summed using MotionCor2 (in 3 x 3 patch mode) [43]. By doing so, dose-weighted and un-weighted full-dose images were generated. Image and data processing were performed with the SPHIRE/EMAN2 software package [44]. Un-weighted full-dose images were used for defocus and astigmatism estimation by CTER [45]. With the help of the drift assessment tool in SPHIRE, drift-corrected micrographs were further sorted to discard high defocus as well as high drift images that could not be compensated for by frame alignment.

For the $PA_7LF_3$ dataset, particles were automatically selected based on a trained model using the crYOLO software, implemented in SPHIRE [46]. In total, 382 k particles were extracted from the dose-weighted full dose images with a final window size of 336 x 336 pixel. Two-dimensional classification was performed using the iterative and stable alignment and clustering (ISAC) algorithm implemented in SPHIRE. Several rounds of 2-D classification yielded a total number of 213 k 'clean' dose-weighted and drift-corrected particles. During the manual inspection of the 2-D class averages, top views of the particles were excluded.

A generated composite crystal structure consisting of $PA_7$ (PDB:1TZO) decorated with three full-length LF (PDB:1J7N), docked with their N-terminal domain to PA as observed in the $PA_8LF_4$ structure (PDB: 3KWV), was converted into electron density (sp_pdb2em functionality in SPHIRE). After filtering to 30 Å, this map served as reference in the initial 3-D refinement. The 3-D refinement without imposed symmetry (sxmeridien in SPHIRE, C1) yielded an initial 3.5 Å electron density map of the $PA_7LF_3$ complex. Several rounds of 3-D classification and rotation of certain classes were necessary to separate particles belonging to $PA_7LF_{2+1A}$, $PA_7LF_{2+1B}$ and $PA_7LF_{(2+1A)'}$ complexes. The respective classes obtained by 3D sorting or from previous 3D refinements served as reference for all subsequent 3D refinements. The flowchart of the image processing strategy including the obtained 3-D classes as well as the number of particles that they contained is described in detail in S3 Fig (S3 Fig).

Global resolutions of the final maps were calculated between two independently refined half maps at the 0.143 FSC criterion, local resolution was calculated using sp_locres in SPHIRE. The final densities were filtered according to local resolution or the local de-noising filter LAFTER was applied to recover features with more signal than noise (based on half maps) [47].

## Model building, refinement and validation

To build the $PA_7$ model, a single monomer of the $PA_8$ crystal structure (PDB: 3HVD) was placed seven times into the density corresponding to the heptameric PA ring of the $PA_7LF_{3-masked}$ map using rigid body fitting in Chimera [48]. Next, two copies of the monomeric lethal factor of the unbound crystal structure (PDB:1J7N) were fitted similarly into the corresponding LF density located atop of the $PA_7$ ring. The resulting model was then flexibly fitted using iMODFIT [49] and subsequently further refined using a combination of manual model building in COOT [50] and real-space refinement in PHENIX [51]. Unresolved loop regions were deleted and side chain information was removed for less well-resolved regions.

The resulting models for $^1LF$, $^2LF$ and $PA_7$ from the $PA_7LF_{3-masked}$ model served again as starting point for the $PA_7LF_{2+1B}$ structure and were placed into the corresponding density using rigid-body fit in Chimera. Chain H ($^1LF$) was duplicated and placed similarly into the density corresponding to the third LF. For the entire model a restrained refinement in PHENIX was performed. The resulting model was further refined using a combination of COOT and real-space refinement in PHENIX. Like for the $PA_7LF_{3-masked}$ model, unresolved loop regions were deleted and side chain information was removed for less well-resolved regions.

The $PA_7LF_{2+1B}$ model served as starting model for $PA_7LF_{2+1A}$ and was initially fitted into the density map using rigid-body fitting. Chain J ($^3LF$) was placed into the density corresponding to the third LF. Similar to $PA_7LF_{2+1B}$, a restrained refinement in PHENIX was performed with the entire model, followed by an additional refinement using a combination of COOT and real-space refinement in PHENIX. Unresolved loop regions were deleted and side chain information was removed for less well-resolved regions. Geometries of the final refined models were obtained either from PHENIX or calculated with the Molprobity server (http://molprobity.biochem.duke.edu). Data statistics are summarized in S1 Table (S1 Table).

## Structure analysis and visualization

UCSF Chimera was used for structure analysis, visualization and figure preparation. The angular distribution plots as well as beautified 2-D class averages were calculated using SPHIRE.

## Supporting information

**S1 Fig. Reconstitution of PA$_7$ into lipid mimetic systems after pore transition.** (**A**) Representative negatively stained electron micrograph areas of PA$_7$ reconstituted into POPC liposomes (top panels), with individual inserted particles highlighted by white arrowheads. Selection of inserted particles in smaller lipid vesicles (bottom panel). Scale bar: 15 nm. Particles are clearly accumulated at lipid membranes. (**B**) Representative negatively stained electron micrograph area of PA$_7$ reconstituted in lipid nanodiscs (MSP1D1), with individual inserted particles highlighted by black arrowheads. Scale bar: 20 nm (**C**) Model of PA$_7$ complexes inserted into lipid nanodiscs with additional examples of individual particles after reconstitution (same nanodiscs as in B). Scale bar: 20 nm.
(TIF)

**S2 Fig. Purification and cryo-EM of PA$_7$LF$_3$.** (**A**) Coomassie-stained SDS-PAGE of purified PA$_7$LF$_3$ complex. (**B**) Size exclusion chromatography profile of the PA$_7$LF$_3$ complex using a Superdex 200 column. Sample fraction used for cryo-EM studies is indicated by black arrow. (**C**) Representative digital micrograph area of vitrified PA$_7$LF$_3$ complex. Scale bar: 20 nm. (**D**) Representative 2-D class averages corresponding to **C**. Scale bar: 10 nm. (**E-H**) Rotated views of the 3-D reconstruction of PA$_7$LF$_{2+1A}$ (**E**), PA$_7$LF$_{(2+1A)'}$ (**F**), PA$_7$LF$_{2+1B}$ (**G**), and PA$_7$LF$_{3-masked}$ (**H**), respectively. (**I**) FSC curves between two independently refined half-maps of PA$_7$LF$_{2+1A}$ (green), PA$_7$LF$_{(2+1A)'}$ (red), PA$_7$LF$_{2+1B}$ (blue) and PA$_7$LF$_{3-masked}$ (purple).
(TIF)

**S3 Fig. Flowchart of image processing strategy in SPHIRE.** The single particle processing workflow is shown that included multiple 3-D classification steps as well as rotation of individual classes (indicated by rotation symbol). Number of particles in each class is provided as orange box below the respective structure and the obtained resolution of the map after 3-D refinement is indicated. For each structure a top and side view is shown (in top views PA$_7$ density is partially clipped to focus on the bound LFs). Mask for masking out third LF is provided in dashed box. Final electron density maps are highlighted by green boxes. Abbreviations: cla3D – 3-D classification, cla2D – 2-D classification, ref-3D – 3-D refinement.
(TIF)

**S4 Fig. Local resolution and 3-D orientation plots.** (**A-D**) Rotated views of the reconstructions, PA$_7$LF$_{2+1A}$ (**A**), PA$_7$LF$_{(2+1A)'}$ (**B**), PA$_7$LF$_{2+1B}$ (**C**), and PA$_7$LF$_{3-masked}$ (**D**), respectively, colored by local resolution. Corresponding color key of local resolution is provided on the right. The position of the N-terminal domain of $^1$LF is indicated by a dashed orange ellipse for orientation. (**E**) Selected examples of side chain densities corresponding to PA and LF with atomic models fitted. (**F**) Rotated views of the 3-D angular distribution plot for the PA$_7$LF$_{2+1A}$ reconstruction, in which the relative height of bars represents the number of containing particles. Corresponding 2-D histogram is shown on the right. (**G-I**) Same as in **F** for PA$_7$LF$_{(2+1A)'}$ (**G**), PA$_7$LF$_{2+1B}$ (**H**), and PA$_7$LF$_{3-masked}$ (**I**).
(TIF)

**S5 Fig. Structure comparison of PAs and LFs.** (**A**) Superposition of the seven PA protomers in PA$_7$LF$_3$, which are colored in different blue hues (left panel), and a single PA subunit (blue) with the known crystal structure (PDB: 1TZO, green, right panel). Loop region 2β2-2β3

(residues 300–323), resolved only in the crystal structure, is highlighted by a black arrowhead. (**B**) Domain organization of LF with individual domains highlighted by different colors. (**C**) Superposition of individual LFs in the $PA_7LF_3$ structures with $^1LF$ in pink, $^2LF$ in gold, $^{3B}LF_N$ in cyan and $^{3A}LF_N$ in dark green. (**D**) Superposition of $^1LF$ (pink), $^2LF$ (gold) and unbound LF (PDB: 1J7N, green), aligned via their C-terminal domain. Green and red arrows indicate similar positions in $^1LF$ and unbound LF (PDB:1J7N), respectively. Comparison reveals that the C-terminal domain is rotated respective to the N-terminal domain in the $PA_7LF_3$ structures. (**E**) Superposition of the N-terminal domain of the three LFs in $PA_7LF_3$ (green), of LF in the "open" conformation in $PA_8LF_4$ (PDB: 3KWV, dark yellow) and of unbound LF in the "closed" conformation (PDB: 1J7N, purple). (**F**) Superposition of the three LF-LF interfaces with $^1LF$-$^2LF$ in pink, $^2LF$-$^{3A}LF$ in green and $^{3B}LF$-$^1LF$ in blue.
(TIF)

**S1 Table. Data collection, refinement and model building statistics.**
(TIF)

**S1 Movie. Conformational change of LF upon PA binding.** The C-terminal domain of the three LF molecules rotates respective to the N-terminal domain upon binding to $PA_7$ when compared with the unbound LF structure (PDB:1J7N), to form a continuous chain of head-to-tail interactions. Top view of the morph between both conformations is shown, with LFs in blue and $PA_7$ in transparent grey.
(MP4)

**S2 Movie. LFs can interact via their C-terminal domains.** In our $PA_7LF_{(2+1A)'}$ reconstruction, two LF molecules interact with each other via their C-terminal domain close to the central axis of the complex, thus forming an additional LF-LF interface. Top view of the morph between this conformation (light blue) and the one observed in the $PA_7LF_{2+1A}$ (yellow) is shown. Volumes are low-pass filtered and shown at the same threshold.
(MP4)

## Acknowledgments

We thank O. Hofnagel and D. Prumbaum for assistance with data collection.

## Author Contributions

**Conceptualization:** Stefan Raunser.

**Data curation:** Claudia Antoni.

**Formal analysis:** Claudia Antoni, Dennis Quentin, Christos Gatsogiannis.

**Funding acquisition:** Stefan Raunser.

**Project administration:** Stefan Raunser.

**Resources:** Alexander E. Lang, Klaus Aktories.

**Supervision:** Christos Gatsogiannis, Stefan Raunser.

**Validation:** Dennis Quentin, Christos Gatsogiannis.

**Visualization:** Claudia Antoni, Dennis Quentin.

**Writing – original draft:** Dennis Quentin, Stefan Raunser.

**Writing – review & editing:** Dennis Quentin, Stefan Raunser.

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
