## [Decision Letter · Decision Letter 0]

16 May 2020

Dear Prof Raunser,

Thank you very much for submitting your manuscript "Cryo-EM structure of the fully-loaded asymmetric anthrax lethal toxin in its heptameric pre-pore state" for consideration at PLOS Pathogens. As with all papers reviewed by the journal, your manuscript was reviewed by members of the editorial board and by several independent reviewers. The reviewers appreciated the attention to an important topic. Based on the reviews, we are likely to accept this manuscript for publication, providing that you modify the manuscript according to the review recommendations.

From the editor: The reviewers agreed the works was of high quality. Two of the referees requested additional discussion to clarify a few points in the context of the current literature. Reviewer #3 in particular provides specific guidance for the content that can address the impact of the work. Please address also the minor points raised by all reviewers regarding text revisions, additional references, and figure presentations. The reviewers all agreed that the criticisms were minor and no additional experiments will be required.

Sincerely,

Karla J.F. Satchell, Ph.D.

Associate Editor

PLOS Pathogens

Michael Wessels

Section Editor

PLOS Pathogens

Kasturi Haldar

Editor-in-Chief

PLOS Pathogens

orcid.org/0000-0001-5065-158X

Michael Malim

Editor-in-Chief

PLOS Pathogens

orcid.org/0000-0002-7699-2064

Reviewer Comments (if any, and for reference):

Reviewer's Responses to Questions

**Part I - Summary**

Reviewer #1: Please see review

Reviewer #2: The manuscript by Antoni and collaborators presents the structure of the anthrax toxin. This complex is constituted of three components: the protective antigen (PA) who forms a multimeric complex that attaches to the TEM8 receptor and two cytotoxic subunits, lethal factor (LF) and the edema factor (EF) which are further translocated into the cytoplasm. The cipher for the geometric assembly of three copies of LF on top of seven copies of PAs was a structural prize followed by many groups over almost two decades.

The present study offers a clear description of the possible distributions of the LFs on the PA ring by calculating a series of cryo-EM maps obtained by sequential 3D classification. Their higher resolution structure shows the LFs arranged in a continuous chain of head-to-tail interactions. However, each of the LF adopt a different conformations which leads the authors to propose a model for the formation of the LFs arrangement and for their translocation.

The manuscript is well constructed, the figures are elegant, clear and informative. I am convinced that the article will be of special interest to readers of PLoS Pathogens and I enthusiastically suggest its publication.

Reviewer #3: This manuscript by Antoni and co-workers use high resolution cryo-EM to study anthrax toxin action on host cells. In particular, the authors seek to gain insights into recruitment of lethal factor and edema factor to the heptameric PA pre-pore and subsequent order of translocation.

The author’s work confirm previous findings that three LFs interact with the heptameric PA to form the loaded pre-pore. However, details of these interactions have been limited due to lower resolution structures. In this study, the authors present higher resolution PA-LF complexes, which offer additional details by which to refine the model of LF and EF recruitment to the PA pre-pore.

The studies described in the present study build upon this previous work by providing higher resolution details. Specifically, although the three LF monomers form a continuous chain of head-to-tail “chain” interactions of with the heptameric PA pre-pore state (PA7LF3), the fate of the three monomers differ, in that while all three monomers are not in the “closed “state, one of the three LF monomers adopts an open conformation while interacting with the alpha clamp of PA, while the other two monomers are in what the authors refer to as an “intermediate” state. Because the third LF can bind the PA7-LF2 complex in two fundamentally different ways, the authors appropriately indicate that they cannot predict the translocation order of monomer

Based on the structures the authors propose a model that differs slightly than that recently proposed by another group, which had predicted that LF translocation occurs by a “chain-reaction” mechanism. Here the authors suggest an alternative to the “chain-reaction” model, and predict that, while the first LF to bind the PA7 complex is likely to be the first translocated substrate, that either the second or third LF could be translocated next, perhaps on differences in the binding orientation of the final recruited LF.

Strengths of the manuscript include the importance of the topic, the high quality of the work, and the well written text. Potential issues include similarities to recently published work from another group, which raises questions about the extent to which the results advance the field.

Although the author’s structural data are convincing and compelling, readers would greatly benefit from a revised discussion where the author’s lay out the evidence in the literature that would support the biological relevance of their findings. In particular, these and other structural studies have relied on interactions between PA and LF at micromolar or near micromolar concentrations which may be unlikely to occur during in vivo infection. What is the existing evidence that, in vivo, three substrates bind to the PA pore at nM or pM concentrations? Please discuss existing quantitative binding that indicates the affinity of these interactions. In addition, the authors should discuss the direct experimental evidence that LF (or EF) are actually translocated when all three sites of PA are occupied? The authors should discuss existing data that indicate that all three PA sites be occupied for translocation, or do existing data indicate that LF/EF can be translocated regardless of whether 1, 2, or 3 effector binding sites on PA are occupied? What do existing data tell us about whether the second or third LF/EF to bind PA are actually ever translocated following translocation of the first molecule? In other words, what DO we currently know, and what do we NOT currently know about the items discussed above in relation to infection in vivo.

**Part II – Major Issues: Key Experiments Required for Acceptance**

Reviewer #1: (No Response)

Reviewer #2: None

Reviewer #3: (No Response)

**Part III – Minor Issues: Editorial and Data Presentation Modifications**

Reviewer #1: Please see Review

Reviewer #2: Minor points:

• line 275 and others: space between the number and symbol

• line 304: mention octyl glucoside while introducing the OG

• line 324: please give the duration of the blotting step

• line 345: please provide reference for CTER

• line 374: reference for Chimera

• line 377: references for COOT and PHENIX

Reviewer #3: Although the author’s structural data are convincing and compelling, readers would greatly benefit from a revised discussion where the author’s lay out the evidence in the literature that would support the biological relevance of their findings. In particular, these and other structural studies have relied on interactions between PA and LF at micromolar or near micromolar concentrations which may be unlikely to occur during in vivo infection. What is the existing evidence that, in vivo, three substrates bind to the PA pore at nM or pM concentrations? Please discuss existing quantitative binding that indicates the affinity of these interactions. In addition, the authors should discuss the direct experimental evidence that LF (or EF) are actually translocated when all three sites of PA are occupied? The authors should discuss existing data that indicate that all three PA sites be occupied for translocation, or do existing data indicate that LF/EF can be translocated regardless of whether 1, 2, or 3 effector binding sites on PA are occupied? What do existing data tell us about whether the second or third LF/EF to bind PA are actually ever translocated following translocation of the first molecule? In other words, what DO we currently know, and what do we NOT currently know about the items discussed above in relation to infection in vivo.

PLOS authors have the option to publish the peer review history of their article (what does this mean?). If published, this will include your full peer review and any attached files.

Reviewer #1: No

Reviewer #2: Yes: Mihnea Bostina

Reviewer #3: No
---

## [Editor Report · Decision Letter 1]

27 Jun 2020

Dear Prof Raunser,

We are pleased to inform you that your manuscript 'Cryo-EM structure of the fully-loaded asymmetric anthrax lethal toxin in its heptameric pre-pore state' has been provisionally accepted for publication in PLOS Pathogens.

Best regards,

Karla J.F. Satchell, Ph.D.

Associate Editor

PLOS Pathogens

Michael Wessels

Section Editor

PLOS Pathogens

Kasturi Haldar

Editor-in-Chief

PLOS Pathogens

orcid.org/0000-0001-5065-158X

Michael Malim

Editor-in-Chief

PLOS Pathogens

orcid.org/0000-0002-7699-2064
---

## [Editor Report · Acceptance letter]

4 Aug 2020

Dear Prof Raunser,

We are delighted to inform you that your manuscript, "Cryo-EM structure of the fully-loaded asymmetric anthrax lethal toxin in its heptameric pre-pore state," has been formally accepted for publication in PLOS Pathogens.

Best regards,

Kasturi Haldar

Editor-in-Chief

PLOS Pathogens

orcid.org/0000-0001-5065-158X

Michael Malim

Editor-in-Chief

PLOS Pathogens

orcid.org/0000-0002-7699-2064